# Investigating the Metabolic Effects of Ultrasound-Induced Lipolysis

**DOI:** 10.3390/ijms26178689

**Published:** 2025-09-06

**Authors:** Jacob Warner-Palacio, Zackery Paxton, Alexis Hassiak, Spencer Willardson, Dustin Edmonds, Luke Sanders, Parker Feltner, Noah Schultz, Christina Nelson, Kyle B. Bills, David W. Sant

**Affiliations:** Department of Research, Noorda College of Osteopathic Medicine, Provo, UT 84606, USA; do27.jswarner@noordacom.org (J.W.-P.);

**Keywords:** ultrasound therapy, high-frequency ultrasound, low-frequency ultrasound, lipolysis, lipid metabolism, non-invasive body contouring, adjunctive therapies

## Abstract

Ultrasound therapy has emerged as a promising non-invasive approach for fat reduction with the potential to improve metabolic health with both high-frequency (1–3 MHz) and low-frequency (35–300 kHz) ultrasound receiving FDA approval for waist circumference reduction between 2010 and 2014. This literature review aims to investigate the current state of research regarding the physiological mechanisms underlying ultrasound-induced lipolysis and lipid mobilization. While many studies focus on the esthetic benefits of lipolytic ultrasound, less attention has been given to the metabolic fate of liberated lipids and whether therapeutic ultrasound can influence systemic health. A literature search was conducted using PubMed to identify clinical trials and mechanistic studies on ultrasound-based fat reduction, with emphasis on peer-reviewed articles published within the past five years. Reported results show average waist circumference reductions of 0.5–3.12 inches and modest weight loss of 0.47–2.5 pounds following three treatment sessions. Existing literature suggests that lipid byproducts may enter systemic circulation, and adjunctive therapies such as lymphatic drainage and aerobic exercise may support their clearance or oxidation. However, studies directly investigating post-lipolytic metabolism are limited. This review synthesizes current findings and proposed mechanisms; and highlights the need for further investigation into the metabolic consequences of ultrasound-induced lipolysis.

## 1. Introduction

With the prevalence of adult obesity rising from 30.5% in 2000 to 41.9% in 2020, ultrasound therapy has gained prominence as a non-invasive method for targeted fat reduction [1,2]. Current FDA approvals are for cosmetic indications alone [3]. However, beyond esthetic applications, non-invasive fat reduction holds potential for improving metabolic health, particularly regarding visceral and mesenteric fat—both of which are linked to insulin resistance and glucose tolerance [4]. This literature review aims to examine the mechanisms of ultrasound-induced lipolysis and summarize current FDA-approved ultrasound therapies. It compares the effectiveness of High-Frequency and Low-Frequency ultrasound and evaluates proposed physiologic mechanisms such as mechanical cavitation, adipocyte disruption, and subsequent lipid mobilization. Furthermore, it explores potential adjunctive treatments and their proposed roles in enhancing lipid clearance and oxidation.

While prior studies report modest reductions in waist circumference and fat volume, with waist circumference reports ranging from 0.5 to 3.12 inches, and some reporting weight loss of 0.47–2.5 pounds, few have addressed the metabolic fate of liberated lipids [3,5]. It remains unclear whether these lipids are oxidized for energy, re-esterified into new fat depots, or excreted [6]. Moreover, metabolic markers such as triglycerides, apolipoproteins, or basal metabolic rate (BMR) changes are rarely assessed in clinical trials. This knowledge gap limits our ability to determine whether ultrasound therapies can meaningfully influence systemic metabolic health outside of esthetic benefits. For both esthetic and metabolic treatments, non-invasive fat reduction has significant potential for improving health outcomes [4]. However, understanding the direct mechanism of action of lipid clearance and metabolic/systemic effects will help target therapies to the most beneficial outcomes for patient treatment. Therapeutic ultrasound may represent an important tool in addressing metabolic health and reducing the downstream effects of obesity [7].

## 2. Methods

In creating this narrative, the researchers conducted a literature review to synthesize both mechanistic and clinical evidence regarding ultrasound-induced lipolysis. In this review, the term “lipolysis” will encompass both physiologic and device-induced breakdown of lipid stores. In classical biochemistry, lipolysis refers specifically to the enzymatic hydrolysis of triglycerides within adipocytes, yielding glycerol and free fatty acids (FFAs) for downstream oxidation, often termed β-oxidation, or re-esterification. However, in the esthetic and therapeutic ultrasound literature, the term “lipolysis” is conventionally applied to describe adipocyte disruption, whether from mechanical cavitation from ultrasound, or referring to the biological process. While this is mechanistically distinct from enzymatic β-oxidation, the resulting effect of releasing intracellular lipids into the interstitial space becomes relatively analogous in outcome, resulting in the “lysis of lipid stores”. As such, the term “lipolysis” has become widely adopted in both clinical trials and regulatory device descriptions [3,8,9,10]. For clarity, throughout this review, the term “lipolysis” refers to the broader sense of lipid disruption or release. Rather than referring to the distinction between ultrasound-induced cavitation and/or thermal injury and specifying β-oxidation or metabolic lipolysis, this follows the field’s established convention. Additionally, “lipolysis” was included as a key search term in the literature review following the established norms of the esthetic medicine field.

The literature search was executed in February 2025 using PubMed/MEDLINE https://pubmed.ncbi.nlm.nih.gov/ (accessed on 17 February 2025) with the primary aim of identifying peer-reviewed human clinical trials and mechanistic studies of ultrasound-based fat reduction. Because ultrasound devices for esthetic fat reduction first achieved FDA 510(k) clearance in the early 2010s with companies such as LipoSonix (Solta Medical, Bothell, WA, USA) and UltraShape (UltraShape Ltd, Livingston, NJ, USA) [8,9,10,11,12,13], while newer combination devices such as Ultimate Contour (Ideal Curves LLC, Lehi, UT, USA) were cleared by 2020 [14,15]. Consequently, the reviewers emphasized studies published 2020-present to capture contemporary devices, applications, designs, and protocols, while deliberately including some earlier pivotal mechanistic trials to compare the progression and development of the field.

Search terms combined controlled vocabulary and free text for “ultrasound”, “adipose”, “lipolysis”, “cavitation”, “therapeutic ultrasound”, “alternative ultrasound use”, “ultrasound-induced lipolysis”, “Ultrasound-induced fat loss”, “ultrasound-treated adipose tissue”, and “metabolic endpoints”. Combined topic terms included “Esthetic medicine AND ultrasound”, “Fat reduction AND ultrasound”, “Lipolysis OR fat reduction”, “Lipolysis OR Adipose”, “Cavitation OR HIFU OR LFU OR HFU”, “Low-Frequency Ultrasound AND Lipolysis”, and “Metabol* OR Oxidation OR Triglyceride AND Ultrasound OR Lipolysis”. Inclusion/exclusion rules were pre-specified to include primarily human clinical trials, prospective cohorts, randomized controlled trials, and case series with *n* > 10 for clinical effect synthesis. Some mechanistic and animal/ex vivo studies were included for pathway modeling and safety contexts, but these were included in reviewing the effectiveness in human subjects. Language selection was restricted to English for full-text synthesis. However, where non-English abstracts suggested important data, full translated texts were accessed if available and feasible. Six primary reviewers searched for research articles and discussed inclusion/exclusion criteria.

In addition to PubMed/MEDLINE, the literature search incorporated ClinicalTrials.gov to include registered clinical trials and FDA 510(k) (https://www.fda.gov accessed on 20 March 2025) summaries for ultrasound-based fat reduction devices. These inclusions ensured regulatory and safety data in the review, as well as relevant gray literature to enhance completeness. This approach allowed for the primary inclusion of contemporary studies (2020–2025), while also investigating earlier influential mechanistic and clinical trials that shaped the field. Notable examples include randomized controlled trials examining aerobic exercise augmentation of ultrasound-induced lipolysis [2], narrative reviews summarizing mechanistic and clinical effects of non-invasive body contouring [3,16], pilot studies exploring metabolic outcomes of mesenteric visceral lipectomy [4], and clinical safety assessments of FDA-approved devices [5,11,17,18]. Additionally, foundational mechanistic studies on adipose tissue lipid metabolism and lipolysis were incorporated to contextualize the observed clinical effects [6,7,19,20]. By integrating regulatory documentation, clinical trials, and mechanistic studies, the review ensured a comprehensive synthesis of human and translational evidence to ultrasound-induced lipolysis.

During the review phase, 260 records were screened by title and abstract for relevance to ultrasound-induced lipolysis. Of these, 200 were excluded as not directly relevant to human trials, leaving 60 full-text articles assessed for eligibility. However, 10 additional animal-only articles were included for mechanistic purposes with sample sizes greater than 10. Articles whose sample size was too small, <10 (*n* = 6) were excluded, as well as combined-energy studies without direct ultrasound-specific data (*n* = 5), lack of relevant outcomes (*n* = 3), and those with full text unavailable or language barriers (*n* = 1). The remaining 41 articles were included in the narrative synthesis, including 3 sources from FDA 510(k)s and 2 sources from ClinicalTrials.gov.

## 3. Results

### 3.1. Approved Ultrasound Therapies

Ultrasound therapies have achieved FDA clearance as non-invasive modalities for body contouring, offering both esthetic benefits and potential avenues for broader metabolic applications [16]. Additionally, therapeutic ultrasound has been used in a variety of other uses outside of diagnostic imaging including musculoskeletal treatments/physical therapy, milk duct clearance, bone healing, and wound healing [21,22]. While ultrasound lipolysis is classified as non-invasive, in that it does not require incisions or anesthesia, it is a biologically active process. At the tissue level, acoustic energy disrupts adipocytes through thermal and/or cavitation-based mechanisms, distinguishing it from both passive and non-surgical approaches [3,14,23]. However, for the purposes of this literature review, “therapeutic ultrasound” refers primarily to lipolysis for body contouring and fat reduction.

These many ultrasound therapies leverage a range of frequencies and intensities to deliver energy to adipose tissue with differing mechanisms of action depending on the ultrasound parameters employed. Ultrasound waves interact in a unique fashion with adipose tissue, attenuating the waves rather than scattering or reflecting them—particularly at high frequencies—which influences penetration depth and efficacy [11,24]. As such, therapeutic ultrasound plays an important role in potential treatments for fat reduction and may improve metabolic and systemic health outcomes [25].

The intensity of ultrasound waves plays a critical role in tissue interaction. Diagnostic ultrasound, while operating at similar frequencies (2–15 MHz) to many therapeutic ultrasound treatments, typically utilizes very low intensity (0.01–0.1 W/cm^2^) for imaging purposes [3]. In contrast, therapeutic ultrasound employs much higher intensities, typically ranging from 0.5 to 6.0 W/cm^2^ in clinical body contouring and can range as high as 150 W/cm^2^ in High Intensity Focused Ultrasound (HIFU) systems [3]. These elevated intensities allow for adipocyte disruption while maintaining patient safety through precise targeting and pulsing techniques [17].

### 3.2. Frequency Classifications and Mechanisms

#### 3.2.1. High-Frequency Ultrasound (HFU)

HFU typically ranges from 1 to 3 MHz, with focused energy generating thermal effects that raise local tissue temperatures above 50 °C [26]. HFU devices such as Liposonix (Solta Medical, Bothell, WA, USA), which received FDA approval in 2011, exemplify the clinical application of this modality, producing measurable reductions in subcutaneous fat thickness and waist circumference [17]. HFU typically ranges from 1 to 3 MHz, with focused energy generating thermal effects that raise local tissue temperatures above 50 °C [26]. HFU devices such as Liposonix, which received FDA approval in 2011, exemplify the clinical application of this modality, producing measurable reductions in subcutaneous fat thickness and waist circumference [17]. However, HFU is typically limited to superficial fat layers due to the higher level of attenuation with higher frequencies [27].

Additionally, HFU demonstrates benefits over a wide range of intensities. While treatments such as Liposonix utilize both high frequency (2 MHz) and high intensity (150 W/cm^2^), other forms of HFU such as LIPOcel (Jeisys Medical Inc., Seoul, South Korea), though operating at 2 MHz, employ a lower intensity of 63 to 87 W/cm^2^, which mechanically disrupts adipocytes with minimal thermal effect [27]. Therapeutic ultrasound treatments utilizing both high intensity and high frequency are typically classified as High Intensity Focused Ultrasound (HIFU), while those with high frequency with moderate-to-low intensity simply retain the classification of HFU. Both HFU and HIFU have similar treatment properties yet vary slightly in their mechanisms of lipolysis [27]. While some sources use HFU and HIFU without specifying the intensity, this review will maintain the convention that HFU operates between 1 and 3 MHz, with intensities of 3–6 W/cm^2^, while HIFU operates between 2 and 7 MHz, with intensities up to 150 W/cm^2^ to maintain clarity and consistency across the references included. Details about the specifics of HFU can be found in Table 1. 

#### 3.2.2. Intermediate Frequency Ultrasound (IFU)

IFU operates around 1 MHz and is typically represented by devices such as UltraShape (UltraShape Ltd, Livingston, NJ, USA) and VASER Shape (Sound Surgical Technologies, Louisville, CO, USA), which employ moderately high frequency (0.2 and 1.0 MHz, respectively) and low-intensity (0.5–6.0 W/cm^2^) ultrasound for therapeutic effect [8]. These devices and treatment procedures combine moderate-intensity ultrasound with massage for enhanced lymphatic drainage. IFU can deliver both mechanical and mild thermal effects, providing a balance of penetration and efficacy. Although less studied for adipose destruction than HFU or LFU, IFU has shown biological activity in adjacent fields, such as microalgae and soft tissue, and may offer an ideal compromise between superficial and deep fat targeting [8].

Specifically, IFU facilitates cell membrane disruption and intracellular metabolite extraction in these microalgae. Ultrasound at 1 MHz and moderate intensities caused partial perforation and surface peeling of Tetraselmis suecica cells, enabling the release of lipids and other bioactive compounds without fully destroying cell viability [8]. This controlled mechanical disruption potentially mirrors the cavitation effects desired in adipose tissue applications. These findings demonstrate how IFU may function in therapeutic lipolysis and support exploration in human applications. Details about the specifics of IFU can be found in Table 1. 

#### 3.2.3. Low-Frequency Ultrasound (LFU)

LFU is defined here as operating between 35 and 300 kHz and relies on mechanical cavitation as its primary mechanism of action. LFU minimizes thermal damage and penetrates deeper adipose layers, making it suitable for targeting visceral or fibrous subcutaneous fat [27]. FDA-approved devices like UltraShape (UltraShape Ltd., Livingston, NJ, USA), which was approved by the FDA in 2014, and Ultimate Contour, which was cleared in 2020, utilize LFU to facilitate non-invasive fat reduction and show significant promise for waist circumference reduction.

Although LFU can penetrate more deeply than both HFU and IFU, the energy dissipation increases rapidly with depth, and the controlled intensity parameters prevent thermal or mechanical damage to internal organs [27]. Previous clinical trials have not reported adverse events involving visceral tissue injury, and most studies highlight mild, transient side effects such as erythema or localized tenderness to the treatment area [3,17,28]. While more long-term data on visceral fat applications is warranted, current evidence supports LFU’s safety profile for body contouring purposes when used within FDA-approved guidelines and anatomical limits. Details about the specifics of LFU can be found in Table 1. 

### 3.3. The Science of Lipolysis

Lipolysis, the breakdown of triglycerides within adipocytes, is central to energy metabolism and therapeutic fat reduction strategies. This tightly regulated process involves the enzymatic action of hormone-sensitive lipase (HSL) and adipose triglyceride lipase (ATGL), activated by hormones such as catecholamines [19]. Released free fatty acids (FFAs) are subsequently oxidized via β-oxidation in mitochondria or re-esterified into triglycerides and stored in alternative fat depots [29]. Ultrasound therapy leverages mechanical and thermal energy to disrupt adipocyte membranes, releasing stored lipids into the circulatory system [3]. The resultant surge in circulating lipids raises critical questions regarding their systemic redistribution, oxidation, and elimination, as well as potential side effects like ectopic fat deposition in non-target tissues.

#### 3.3.1. HIFU Lipolysis

The various modalities of ultrasound-induced lipolysis are similar, but each employs a unique mechanism. HIFU utilizes high intensity to generate localized thermal effects, leading to coagulative necrosis of fat cells [8]. While all HIFU is a form of HFU simply by definition of the frequencies employed, not all HFU is delivered with the intensity and focus required to induce coagulative necrosis. As such, distinguishing HFU from HIFU subtypes is critical when evaluating therapeutic intent and physiologic outcomes. This form (HFU) of necrosis preserves tissue architecture but results in irreversible cell death. HIFU tends to localize to more superficial tissues without causing damage to the cell membrane (see Figure 1) [9,30].

Conversely, LFU relies primarily on mechanical cavitation, disrupting the adipocyte membranes without significant thermal damage, ultimately resulting in liquefactive necrosis and cellular fragmentation [8]. Both modalities possess unique strengths, with HIFU providing superficial access and localized necrosis, while LFU offers greater penetration depth, targeting deeper adipose tissue, particularly in visceral or mesenteric fat [3], though with some studies showing variable results depending on the exact frequency, tissue impedance, and treatment duration [31]. However, understanding the mechanisms of lipid clearance, oxidation, metabolism, excretion, or redistribution remains limited (see Figure 2).

HIFU-induced coagulative necrosis involves complex immune-mediated mechanisms. Damage-Associated Molecular Patterns (DAMPs) released from necrotic adipocytes activate resident immune cells, initiating a cascade of inflammation and cellular clearance [25]. Neutrophils and monocytes are recruited to the affected area, followed by macrophages that phagocytize necrotic debris. Early-phase macrophages dominate this process, releasing pro-inflammatory cytokines to clear damaged tissue. Over time, late-stage macrophages transition the immune response toward tissue repair and remodeling, releasing anti-inflammatory mediators [9]. This clearance pathway may result in the accumulation of lipid-laden macrophages or foam cells, resulting in immediate adipose reduction, but potentially causing metabolic concerns [25]. Over time, late-stage macrophages transition the immune response toward tissue repair and remodeling, releasing anti-inflammatory mediators [9]. This clearance pathway may result in the accumulation of lipid-laden macrophages or foam cells, resulting in immediate adipose reduction, but potentially causing metabolic concerns [25]. As previously discussed, details regarding the metabolic pathways of these released lipids remain uncertain, underscoring the need for metabolic profiling in future studies [32].

Clinical observations indicate that HIFU significantly reduces localized fat volumes. Literature suggests that 82% of fat cell removal occurs within the first 12 weeks post-treatment, with 95% clearance by 18 weeks [11,24]. While macrophages play a crucial role in lipid processing, the precise timeline and extent of lipid oxidation versus redistribution are not fully defined. Preliminary findings from ultrasound-based therapies indicate average reductions in waist circumference of up to 3.12 inches, along with modest weight loss of 0.47 pounds after three 20 min sessions [18]. These outcomes suggest the high efficacy of ultrasound in reducing adiposity and fat volume but also indicate that adjunctive measures may be necessary to eliminate mobilized lipids [2].

The induction of coagulative vs. liquefactive necrosis (HIFU vs. LFU) represents a fundamental difference in the downstream metabolic consequences and physiological burden following ultrasound lipolysis. HIFU’s use of high-intensity thermal energy leads to coagulative necrosis, where adipocytes die in place and initiate an immune-driven clearance process [9,26]. This places a substantial burden on the immune system, which must respond by recruiting neutrophils and macrophages to the affected area [25]. As macrophages engulf necrotic debris, some become lipid-laden or transform into foam cells—a phenomenon typically associated with atherogenesis and chronic inflammation [32]. Over time, these macrophages either contribute to localized inflammation or must be cleared via the lymphatic system, but this pathway is inefficient and highly variable [24]. The immune system’s reliance on macrophage-mediated clearance raises questions about long-term metabolic effects, including low-grade inflammation, delayed clearance, or unwanted deposition of residual lipids in ectopic tissues [11,25,32]. Over time, these macrophages either contribute to localized inflammation or must be cleared via the lymphatic system, but this pathway is inefficient and highly variable [24]. The immune system’s reliance on macrophage-mediated clearance raises questions about long-term metabolic effects, including low-grade inflammation, delayed clearance, or unwanted deposition of residual lipids in ectopic tissues [11,25].

#### 3.3.2. LFU Lipolysis

In contrast, LFU induces liquefactive necrosis via mechanical cavitation, resulting in direct adipocyte membrane rupture and the immediate release of lipids into the interstitial fluid and bloodstream [3,8]. This mode of action shifts the burden of lipid processing from immune cells to metabolic pathways. Once in circulation, free fatty acids (FFAs) and glycerol bind to carrier molecules such as albumin and apolipoproteins, are transported to the liver, and can be oxidized for energy via β-oxidation or re-esterified into triglycerides [33]. The presence of elevated FFAs may transiently increase metabolic rate and oxygen consumption, especially if coupled with aerobic activity or lymphatic stimulation [2].

Additionally, this pathway may have a more predictable metabolic outcome, provided the liver and peripheral tissues efficiently process the mobilized lipids. However, without timely oxidation or excretion, LFU-mobilized lipids could still be stored in unintended depots [32]. Despite this potential benefit, some literature indicate a theoretical risk of hepatic overload, potentially through hepatic steatosis or insulin resistance with excess circulating FFAs if poorly managed, though these claims remain speculative [34]. Understanding the balance between these outcomes is critical to optimizing ultrasound therapy and highlights the need for further studies using lipidomics and metabolic tracking to clarify the fate of ultrasound-released fat.

### 3.4. Therapeutic Ultrasound Applications and Limitations

FDA-approved ultrasound therapies are considered safe and effective for circumferential waist reduction and body contouring, but they are not approved for obesity treatment or metabolic disease management [11,16,35]. Treatments with ultrasound are deemed “non-surgical” and “non-invasive”, and while they maintain these designators, they are biologically active, posing a potential risk to metabolic health [14,23,36]. However, despite the potential effect on metabolism, the primary treatment remains in esthetics [4]. Frequency-specific protocols, appropriate patient selection, and integration of adjunctive therapies may optimize clinical success for metabolic improvements with treatment.

#### 3.4.1. Safety Considerations and Current Evidence

Safety data in clinical trials consistently report minimal adverse events. HIFU treatments typically result in waist circumference reduction with transient discomfort. Among reported adverse events are as follows: ringing in the ears, discomfort to heat of the ultrasound device, discomfort to tissue texture changes, psychological discomfort regarding waist tissue changes [3,14,17,23,24]. Examples of specific adverse events as reported by Liposonix (*n* = 60) in 2013 include the following: Ecchymosis (11.67%), Tenderness (30%), Dysesthesia (5%), Itchiness (5%), Discoloration (8.33%), with the absence of any significant pain, discomfort, or reported changes on follow up after 20 weeks [12,13,17].

However, the strength of current evidence is limited by small sample sizes, short-term follow up (generally >24 weeks), and a predominant focus on esthetic outcomes rather than mechanistic endpoints such as BMR, lipidomic profiling, insulin sensitivity, lipid profiles, and CBC changes [3,17,24,37]. Long-term safety concerns such as chronic inflammation, ectopic deposition, and adverse metabolic shifts remain theoretical due to a lack of longitudinal data and mechanistic studies into metabolic effects. Future research could incorporate standardized adverse event protocols, metabolic assessments, extended monitoring of biomarkers, and histological or imaging-based tracking of lipid clearance dynamics to establish comprehensive safety and efficacy profiles beyond simply cosmetic evaluations.

#### 3.4.2. Current Hypotheses on Lipid Fate

Several hypotheses have been articulated regarding the fate of liberated FFAs. First, oxidative metabolism, as previously discussed, may collect released FFAs into hepatocytes for subsequent β-oxidation, potentially altering the primary source of energy production from glycolysis to lipolysis [2,38,39]. This hypothesis implies that the liberation of FFAs into the lymphatic system and bloodstream induces a change in the primary metabolic targets of treated individuals, just as when exercise stimulates systemic energy demand, increasing FFA utilization. Supporting this, studies in individuals with fatty liver disease demonstrate that exercise increases hepatic FFA fluxes approximately two-fold, while also enhancing utilization of VLDL-triglycerides, which may indicate an upregulation of lipid oxidation pathways [38]. Additionally, previous results may indicate that myocardial FFA oxidation increased during exercise in MAFLD participants but decreased in controls, suggesting tissue-specific shifts in FFA handling under metabolic stress [38,39].

Second, in energy-replete states, or in situations where metabolic demand is insufficient, the liberated FFAs may undergo re-esterification and storage. This hypothesis posits that existing adipose depots may collect the freed lipids, simply re-depositing the lipid stores in other areas of the body [29]. These in vitro studies have shown that basal re-esterification mediated by DGAT1/2 serves a protective role, directing liberated fatty acids into triglyceride resynthesis and buffering mitochondrial FA overload. Additionally, pharmacological inhibition of these enzymes redirected FFAs toward increased mitochondrial respiration [29]. Consequently, it may be inferred that the body maintains protective factors against liberated FFAs, and due to limitations of metabolism based on energy demand, these FFAs may not be immediately utilized for energy needs, and simply re-deposit in other depots.

Finally, following ultrasound-induced lipolysis, an additional hypothesis posits that an inflammatory response facilitates the clearance of lipid debris through macrophage infiltration and phagocytosis. The damaged adipocytes attract macrophages to form crown-like structures (CLSs) which will engulf lipid droplets and cellular debris [14,23,36]. This process leads to the formation of lipid-laden, or “foamy” macrophages, often termed foam cells that serve as immune mediators of cellular debris rather than the re-esterification process or metabolic oxidation [14,23,36]. Histological studies evaluating HIFU-treated tissue reveal that macrophages can remove disrupted adipocyte components, which enables normal healing without direct metabolism [14,36]. Beyond this, research on adipose tissue macrophages highlights how these lipid-engorged cells adopt distinct metabolic activation profiles, suggesting they may function as transient lipid reservoirs, buffering local fatty acid overload through lysosomal processing rather than typical fat metabolism [14,23,36].

Because of the varying hypotheses, and multiple outcomes potentially observed in various settings, it becomes very likely that a combination of these hypotheses may make up the metabolic fate of these freed fatty acids. Due to the lack of standardization of procedures, and lack of literature investigating the long-term metabolic effects of liberated fatty acids, mechanistic studies may represent a promising area of research in the developing field. Further identifying the interplay of these hypothetical outcomes to liberated lipids will help elucidate the end fate of these body-contouring treatment methods.

#### 3.4.3. Standardization of Treatment Methods

In part due to the lack of standardization in treatment methods, many forms of therapeutic ultrasound rely on adjunctive therapies for treatment, which include massage, lymphatic drainage techniques, radiofrequency, and red-light therapy [32]. Typically, HFU relies on skin tightening treatments (radio and red light), while LFU often benefits from post- and during-treatment lymphatic massage [8,32]. These multimodal approaches may help bridge the gap between esthetic and functional medical outcomes, allowing targeted approaches to induce lipolysis in the areas most beneficial for therapeutic outcomes.

Furthermore, adjunctive treatments such as aerobic exercise and osteopathic lymphatic techniques may play a pivotal role in enhancing lipid clearance and preventing ectopic fat deposition [40,41]. Following ultrasound-induced lipolysis, free fatty acids (FFAs) and glycerol enter systemic circulation and require prompt metabolism to prevent re-esterification into adipose tissue [20]. Despite these promising adjunctive strategies, there is still a lack of robust clinical data linking ultrasound-induced fat disruption to sustained metabolic improvements.

In addition to the lack of standardization in treatment methods, procedures, and adjunctive therapies, few reviews highlight the targeted mechanism of action in therapeutic ultrasound. Several studies emphasize cosmetic endpoints such as waist circumference or patient satisfaction, without evaluating changes in basal metabolic rate (BMR), lipid profiles, or insulin sensitivity [11,24]. This lack of emphasis on mechanistic underpinnings leads to a significant gap in clinical literature, as many FDA-approved studies tend to focus on esthetic improvements rather than metabolic changes and clinical outcomes [3].

### 3.5. Complimentary Treatments with Ultrasound

Adjunctive therapies and complementary interventions represent a wide range of additive treatments. Physical adjuncts such as lymphatic massage, Osteopathic Manipulative Treatment (OMT), and physical therapies aim to increase drainage and improve mobilization [10,32]. Lifestyle interventions including aerobic exercise immediately following treatment and dietary changes aim to alter metabolic rate and tendencies [2,38,39]. Additionally, technological interventions including radiofrequency, red-light therapy, and infrared aim to affect local metabolites, skin turgor, and increase efficacy [3,16,37]. These strategies may significantly enhance the clinical efficacy of ultrasound-induced lipolysis [2,25]. When adipocytes are disrupted, whether thermally through HIFU or mechanically via LFU, the resulting release of FFAs and glycerol into circulation necessitates timely metabolic processing to avoid re-esterification and ectopic fat deposition [20]. Taken together, complementary strategies broaden the therapeutic scope of ultrasound lipolysis, ensuring that treatment addresses not only local fat disruption but also effective clearance and long-term esthetic and metabolic goals. Adjunctive therapies and complementary interventions represent a wide range of additive treatments. Physical adjuncts such as lymphatic massage, Osteopathic Manipulative Treatment (OMT), and physical therapies aim to increase drainage and improve mobilization [10,32]. Lifestyle interventions including aerobic exercise immediately following treatment and dietary changes aim to alter metabolic rate and tendencies [2,38,39]. Additionally, technological interventions including radiofrequency, red-light therapy, and infrared aim to affect local metabolites, skin turgor, and increase efficacy [3,16,37]. These strategies may significantly enhance the clinical efficacy of ultrasound-induced lipolysis [2,25]. When adipocytes are disrupted, whether thermally through HIFU or mechanically via LFU, the resulting release of FFAs and glycerol into circulation necessitates timely metabolic processing to avoid re-esterification and ectopic fat deposition [20]. Taken together, complementary strategies broaden the therapeutic scope of ultrasound lipolysis, ensuring that treatment addresses not only local fat disruption but also effective clearance and long-term esthetic and metabolic goals.

#### 3.5.1. Physical Adjuncts

Physical adjunctive therapies provide direct mechanical stimulation and manual support to aid the body’s natural clearance systems, complementing ultrasound by improving circulation, reducing congestion, and aiding in the post-treatment recovery. Specifically, by incorporating OMT into the post-treatment plan, lymphatic and venous return can be augmented through rib-raising, thoracic pump, pedal pump, targeted effleurage, and targeting diaphragmatic restrictions [10]. These interventions may accelerate clearance of debris and reduce edema, synergizing with ultrasound’s lipolytic effects. Similarly, lymphatic massage and manual lymphatic drainage are widely applied in cosmetic protocols and have been proposed to accelerate the mobilization of lipids through lymphatic channels, reducing the likelihood of redeposition into ectopic depots [32,37]. In obesity, lymphatic dysfunction contributes to impaired lipid trafficking, and mechanical stimulation restoring lymphatic patency may be particularly beneficial in increasing mobilization [40,41].

Moreover, clinical reports suggest that massage applied after (or during) LFU-based treatments reduces discomfort, expedites recovery, and enhances patient-perceived contour changes [3]. These measures also support immune clearance of apoptotic adipocytes, potentially reducing local inflammation [23]. Because of these initial findings, physical adjuncts may provide a supportive framework for enhancing the body’s systemic mechanisms while reinforcing the localized benefits of ultrasound therapy.

#### 3.5.2. Lifestyle Interventions

Lifestyle interventions represent low-cost but high-value strategies to optimize ultrasound outcomes. Aerobic exercise is a particularly effective complement and can be readily paired with ultrasound therapy treatments. Studies in metabolic dysfunction-associated fatty liver disease (MAFLD) demonstrate that exercise increases hepatic FFA oxidation and enhances utilization of VLDL-triglycerides, shifting systemic fuel use toward lipids [2,38,39]. Even modest activities such as brisk walking or cycling after treatment can bias released lipids toward oxidation rather than storage, potentially reducing risks of ectopic fat deposition. The timing of these interventions may prove essential, and initiating exercise in the hours following ultrasound could synchronize mobilization with oxidative demand.

Dietary interventions are another promising adjunct. Restricting carbohydrate intake in the immediate post-treatment window may reduce insulin-mediated suppression of lipolysis and promote FFA oxidation [34]. Similarly, adequate hydration supports lymphatic transport and may facilitate clearance of cellular debris. Together, these interventions create a metabolic environment favorable for lipid utilization, aligning acute energy demands with ultrasound-induced lipolysis and reinforcing the synergy between procedural and lifestyle measures.

#### 3.5.3. Supplemental Treatments or Metabolic Activation

The choice of technology also plays a role in determining the best complementary strategy and may include additional technologies to mitigate adverse effects of ultrasound treatments. HIFU induces coagulative necrosis, resulting in both fat cell death and subsequent skin tightening via collagen contraction and remodeling [9]. This dual effect makes HIFU particularly suitable for patients who desire both adipose reduction and dermal rejuvenation. In contrast, LFU’s cavitational mechanism better targets fibrous or deeper adipose compartments where collagen stimulation is less relevant [25]. Thus, a tailored approach—potentially combining HIFU for dermal tightening with LFU for deeper fat disruption—can optimize outcomes based on fat distribution and treatment goals.

Beyond ultrasound itself, technological adjuncts such as radiofrequency (RF) and photobiomodulation (PBM) extend treatment potential. RF generates resistive heating in the dermis, promoting collagen remodeling and modest adipocyte injury; when paired with ultrasound, it can enhance skin tightening and improve contour uniformity [16,28,32]. PBM using red and near-infrared wavelengths has been shown to increase adipocyte membrane permeability and may accelerate lipid release when combined with ultrasound [3,16,37]. Although clinical data for PBM as a stand-alone fat reduction tool are mixed, evidence supports its utility as an adjunct to lipolytic devices, particularly for patients prioritizing circumference reduction and skin texture.

#### 3.5.4. Integrated Multimodal Protocols

Optimizing outcomes from ultrasound therapy requires more than simple energy delivery. Multimodal protocols that incorporate physical therapies, metabolic activation, and technological supplements may maximize both esthetic and metabolic benefits. Programs that sequence HIFU with RF can simultaneously decrease fat stores and improve dermal tone, and pairing with PBM and radiofrequency/Red-light therapy may reduce inflammation and accelerate healing [16,37]. Embedding aerobic exercise prescriptions into the treatment course may also ensure that liberated lipids are oxidized efficiently rather than immediately re-esterified in distant fat depots [38,39].

In high-risk populations, including patients with MAFLD, insulin resistance, or obesity, these multimodal strategies may be especially important and expand beyond simply esthetic appeal, given the impaired lipid trafficking and clearance common in these populations [40,41]. Standardizing protocols that integrate lifestyle guidance, manual therapies, and technological adjuncts could help ensure reproducible outcomes and establish metabolic and cosmetic safety profiles. These adjunctive approaches can extend the impact of ultrasound by aligning localized adipocyte disruption with systemic physiologic mechanisms to achieve patient-specific goals. This may help reinforce the value of therapeutic ultrasound beyond body contouring and into potential benefits for metabolic health.

## 4. Discussion

This literature review highlights both the clinical and scientific ambiguity surrounding ultrasound-induced lipolysis. While numerous studies confirm that both HIFU and LFU modalities can reduce waist circumference and subcutaneous fat thickness, the metabolic fate of liberated lipids remains poorly understood. These modalities induce distinct forms of necrosis—coagulative for HIFU and liquefactive for LFU—which likely alter clearance pathways and systemic effects. Without standardized protocols for frequency, intensity, or adjunctive interventions, comparing outcomes across studies is difficult, limiting both mechanistic insight and clinical translation.

Despite consistent reports of esthetic improvement, few studies address whether ultrasound-mediated fat loss contributes to meaningful systemic changes such as improved lipid profiles, insulin sensitivity, or basal metabolic rate. Critical biomarkers—including triglycerides, apolipoproteins, FFAs, ketone bodies, and inflammatory mediators—are rarely incorporated into trial designs. Advanced tools such as continuous glucose monitoring (CGM), lipidomics, and VO_2_ max testing could provide a more complete picture of metabolic outcomes. The absence of longitudinal data leaves open whether observed fat reduction represents a transient cosmetic effect or a sustained metabolic benefit, particularly in patients with obesity or metabolic syndrome.

The role of the immune system and clearance mechanisms remains another gap. Foam cell formation, macrophage polarization, and subclinical inflammation following necrosis are seldom quantified, yet may influence long-term outcomes. Similarly, lymphatic transport and hepatic lipid processing after LFU vary across individuals depending on fitness, liver health, and metabolic flexibility. These factors warrant systematic investigation.

Adjunctive therapies—including red light and radiofrequency treatments, exercise, dietary strategies, and lymphatic drainage—are frequently used but rarely standardized. Such approaches likely enhance lipid oxidation and clearance, but their contributions remain difficult to isolate without controlled comparative trials. From an osteopathic perspective, lymphatic techniques (e.g., thoracic inlet release, lymphatic pumps, effleurage) may offer non-invasive, low-risk means of improving clearance and reducing ectopic deposition. Integrating these manual therapies with ultrasound could broaden applications beyond esthetics and align them with holistic, preventive medicine.

We propose that future research adopt standardized protocols that delineate the roles of HIFU and LFU and specify timing, duration, and intensity of adjunctive therapies. Stratifying patients by metabolic status, fat distribution, and cardiovascular fitness would allow more personalized risk-benefit evaluation and improve reproducibility. This framework would also provide the clarity needed for regulatory guidance and safe clinical integration.

To advance this goal, controlled trials should incorporate multimodal monitoring. Key endpoints include lipid panels (HDL, LDL, triglycerides, total cholesterol), FFAs, apolipoproteins, ketone bodies, and inflammatory markers (CRP, IL-6, TNF-α), tracked at baseline, during treatment, and longitudinally. Complementary measures such as CGM, VO_2_ max, resting metabolic rate, and body composition (DXA or MRI) would clarify systemic impact. Randomization into post-treatment activity groups—such as control, OMT, or moderate aerobic exercise—could determine whether physical adjuncts accelerate clearance or improve efficiency. These comprehensive datasets would yield crucial mechanistic insights into lipid fate and define the contribution of adjunctive therapies.

In summary, while ultrasound lipolysis is a validated technique for body contouring, its systemic implications remain underexplored. Priority areas for future research include metabolic fate of lipids, immune and hepatic clearance mechanisms, adjunctive therapy standardization, and osteopathic or lifestyle interventions. A more rigorous, interdisciplinary approach could move ultrasound lipolysis beyond esthetics and position it as a potential adjunct in managing obesity and metabolic dysfunction.

## Figures and Tables

**Figure 1 ijms-26-08689-f001:**
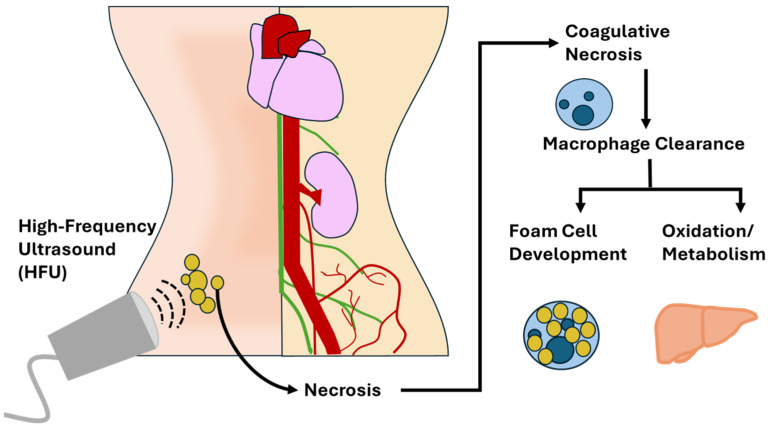
Infographic representing the pathway of processing the fat after treatment with High-Frequency Ultrasound (HFU).

**Figure 2 ijms-26-08689-f002:**
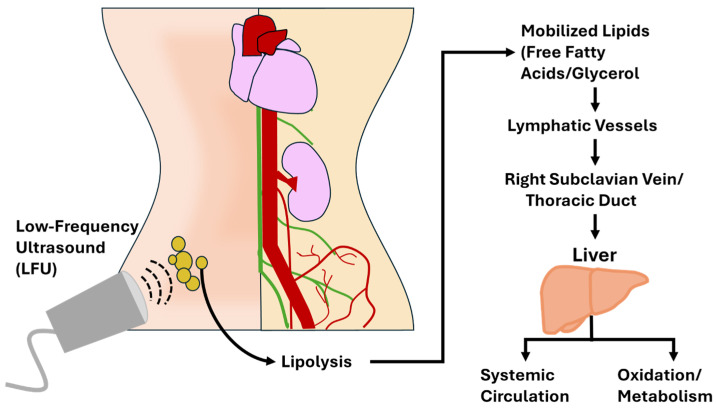
Infographic representing the pathway of processing fat after treatment with Low-Frequency Ultrasound (LFU).

**Table 1 ijms-26-08689-t001:** Information regarding various types of ultrasound treatment.

Feature	HIFU (High- Intensity Focused Ultrasound)	HFU (High- Frequency Ultrasound)	IFU (Intermediate- Frequency Ultrasound)	LFU (Low- Frequency Ultrasound)	Diagnostic Ultrasound
Frequency	2–7 MHz	1–3 MHz	~0.8–1.0 MHz	35–300 kHz	2–15 MHz
Intensity	Very High (up to 150 W/cm^2^ or more)	Moderate (3–6 W/cm^2^)	Low–Moderate (0.5–6 W/cm^2^)	Low–Moderate (1–5 W/cm^2^)	Very Low (0.01–0.1 W/cm^2^)
Primary Mechanism	Thermal coagulative necrosis (focused)	Thermal with some mechanical	Mild thermal & mechanical disruption	Mechanical cavitation	Imaging via echo reflection
Necrosis Type	Coagulative necrosis	Coagulative necrosis (mild)	Mild cell disruption, partial rupture	Liquefactive necrosis	None
Penetration Depth	Superficial (1.5 mm–1.3 cm focal depth)	Shallow to mid-layer (~0.5–1.5 cm)	Mid-depth (1–2 cm)	Deeper (~2–4 cm)	Shallow to moderate
Target Area	Localized fat, subcutaneous/ visceral	Subcutaneous fat	Subcutaneous and some visceral fat	Deep subcutaneous or visceral fat	Internal organ imaging
Lipid Fate	Cleared by macrophages and lymphatics	Partially immune-cleared	Mixed (some circulation, some clearance)	Transported to liver, oxidized, or stored	N/A
Treatment Duration	~30–60 min	~20–30 min	~20–30 min	~15–30 min	Real-time during imaging
Examples (FDA)	Liposonix, Scizer	LIPOcel,	VASER Shape, MedContour	UltraShape, Ultimate Contour	All standard US machines
Adjunct Therapies	RF, red light, lymphatic drainage	RF, massage	Massage, lymphatic drainage	Lymphatic drainage, aerobic exercise	N/A

## Data Availability

No new data were created or analyzed in this study. Data sharing is not applicable to this article.

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
