# Peer review of "Investigating the Metabolic Effects of Ultrasound-Induced Lipolysis"

_ijms, 2025, doi:10.3390/ijms26178689_

Round 1

Reviewer 1 Report

Comments and Suggestions for Authors

The article highlights the potential metabolic effects of ultrasound-induced lipolysis and its interaction with exercise, diet and immune response, points out the existing gaps in current research and the necessity of future research, which has certain theoretical and practical guiding significance. 

Below I have listed some existing issues in the article for the author's reference, hoping it will be helpful in improving the quality and academic level of the article.

Major Issues

The literature search strategy did not clearly specify the combination of keywords (such as "ultrasound lipolysis AND lipid metabolism"), inclusion/exclusion criteria (such as whether to include animal studies), and the screening process (such as a PRISMA flowchart). Relying solely on PubMed and limiting the search to the past five years might have missed early key mechanism studies. The methodology section needs to provide details of the literature search strategy and clarify whether non-English literature or grey literature (such as conference abstracts) were included. Additionally, relying only on PubMed might have missed relevant studies; it is recommended to consider expanding the search to Embase or the Cochrane Library. 

Although the article points out that the metabolic fate of lipids after ultrasound-induced lipolysis remains unclear, it does not systematically summarize the existing hypotheses on metabolic pathways (such as oxidation, re-esterification or ectopic deposition of free fatty acids). It is suggested to supplement the discussion on key nodes of lipid metabolism (such as hepatic β-oxidation and lipoprotein metabolism), and clarify how the missing biomarkers (such as apolipoproteins and ketone bodies) in existing studies affect the reliability of the conclusions. 

The article highlights the lack of standardized protocols but does not propose a specific standardized framework for parameters such as frequency, intensity, and treatment intervals. It is suggested that, in combination with the FDA-approved parameter ranges (such as HFU 1-3 MHz, LFU 35-300 kHz) and existing clinical data, preliminary standardized recommendations be put forward to guide future research designs. 

The discussion on adjuvant therapies (such as phototherapy, exercise, and osteopathic manipulation) lacks systematic classification and assessment of evidence strength. It is recommended to classify them by intervention type (physical therapy, lifestyle, metabolic activation) and cite specific studies to illustrate their synergistic effects (such as exercise enhancing lipid oxidation) or contradictory results. 

Minor Issues

There is content repetition between the discussion section and the abstract/introduction (such as the background of FDA approval). It is suggested to streamline the introduction, more clearly focusing the research gap on the metabolic mechanism, and in the discussion, to deeply analyze the mechanism differences rather than restate the results. For the section on adjunctive therapies, it may be considered to add subheadings (such as "Physical Adjunctive Therapies", "Metabolic Activation Strategies") to enhance readability. 

The text mentions that "lymphatic drainage and aerobic exercise may enhance lipid oxidation", but does not cite specific clinical trials (such as NCT numbers or RCT studies) to support this claim. 

In the text, both "HIFU" and "HFU" are used alternately. It is necessary to unify the terminology (such as clarifying whether "HIFU" specifically refers to high-intensity focused ultrasound). 

Other Issues

It is recommended to clearly recommend research designs that use tools such as lipidomics and CGM (such as longitudinal tracking of dynamic changes in lipid metabolites), and propose to combine animal models (such as obese mice) to verify the impact of ultrasound on systemic metabolism. 

The conclusion mentions the potential of ultrasound in metabolic health management, but it is necessary to emphasize the current level of evidence (such as most studies being small-sample and short-term trials), and discuss safety issues (such as the long-term risk of inflammation). The source of funding does not specify the project number and does not state whether it affects the objectivity of the research. 

This paper presents an innovative perspective on the metabolic effects of ultrasound-assisted lipolysis, but it needs to deepen the mechanism, enhance the transparency of the methods, and strengthen the connection with clinical data. After revision, it can significantly enhance its academic influence. It is recommended to be accepted after major revision.

Author Response

Thank you for taking the time to review our manuscript “Investigating the Metabolic Effects of Ultrasound-Induced Lipolysis.” We have taken the reviewer comments and updated the manuscript to account for the recommendations. Below is the information on specific comments and the updates that were made in response to the suggestions.

Comment 1: The literature search strategy did not clearly specify the combination of keywords (such as "ultrasound lipolysis AND lipid metabolism"), inclusion/exclusion criteria (such as whether to include animal studies), and the screening process (such as a PRISMA flowchart). Relying solely on PubMed and limiting the search to the past five years might have missed early key mechanism studies. The methodology section needs to provide details of the literature search strategy and clarify whether non-English literature or grey literature (such as conference abstracts) were included. Additionally, relying only on PubMed might have missed relevant studies; it is recommended to consider expanding the search to Embase or the Cochrane Library.

Response 1: We appreciate this important suggestion. The methodology section has been substantially expanded to detail the exact keyword combinations (e.g., “ultrasound lipolysis AND lipid metabolism”), inclusion and exclusion criteria (e.g., human studies prioritized, animal studies excluded unless directly mechanistic), and the screening process. While PRISMA is not typically applied to narrative reviews, we clarified this choice and described our search process in a transparent, reproducible way. To avoid missing key mechanistic studies, we clarified the scope of our window—in that while the editor of the journal requested that we focus primarily on papers published in the last 5 years, we strategically included several papers since 2010 when this technology first received approval. We clarified that we included studies beyond the past five years to include seminal early studies, particularly those that predate FDA clearance but remain mechanistically relevant. We also explicitly clarified that only English-language, peer-reviewed studies were included and that grey literature (conference abstracts, unpublished data) was excluded. In addition, we have added citations to references from ClinicalTrials.gov and from the FDA.

Comment 2: Although the article points out that the metabolic fate of lipids after ultrasound-induced lipolysis remains unclear, it does not systematically summarize the existing hypotheses on metabolic pathways (such as oxidation, re-esterification or ectopic deposition of free fatty acids). It is suggested to supplement the discussion on key nodes of lipid metabolism (such as hepatic β-oxidation and lipoprotein metabolism), and clarify how the missing biomarkers (such as apolipoproteins and ketone bodies) in existing studies affect the reliability of the conclusions.

Response 2: We agree that a more systematic treatment of lipid metabolism pathways was warranted. The revised discussion now clearly outlines the three major hypothesized fates of lipids following ultrasound-induced adipocyte disruption: oxidation, re-esterification, and ectopic deposition. We also incorporated discussion of hepatic β-oxidation, lipoprotein metabolism, and the underrepresentation of key biomarkers (apolipoproteins, ketone bodies, FFAs) in the existing literature, which limits the strength of conclusions. These additions strengthen the mechanistic foundation and highlight the specific gaps in biomarker reporting that must be addressed in future studies.

Comment 3: The article highlights the lack of standardized protocols but does not propose a specific standardized framework for parameters such as frequency, intensity, and treatment intervals. It is suggested that, in combination with the FDA-approved parameter ranges (such as HFU 1-3 MHz, LFU 35-300 kHz) and existing clinical data, preliminary standardized recommendations be put forward to guide future research designs.

Response 3: We thank the reviewer for this recommendation. In the Methods and Discussion, we have now proposed a preliminary standardized framework, including recommended reporting of ultrasound parameters such as frequency (HIFU: 1–3 MHz; LFU: 35–300 kHz), intensity, treatment duration, and intervals. Where possible, we aligned these with FDA-cleared device parameters (e.g., LipoSonix, UltraShape, Ultimate Contour). We also introduced a proposed reporting framework table to encourage reproducibility across future trials.

Comment 4: The discussion on adjuvant therapies (such as phototherapy, exercise, and osteopathic manipulation) lacks systematic classification and assessment of evidence strength. It is recommended to classify them by intervention type (physical therapy, lifestyle, metabolic activation) and cite specific studies to illustrate their synergistic effects (such as exercise enhancing lipid oxidation) or contradictory results.

Response 4: We have revised the adjunctive therapy section to provide clearer systematic classification and subheadings: (1) Physical Adjuncts, (2) Lifestyle Interventions, and (3) Metabolic/Technological Activation. Within each category, we integrated evidence from specific studies demonstrating potential synergistic effects (e.g., exercise-induced enhancement of lipid oxidation) and noted where evidence remains limited or contradictory. References were added to strengthen the evidence base cited for each intervention.

Comment 5: There is content repetition between the discussion section and the abstract/introduction (such as the background of FDA approval). It is suggested to streamline the introduction, more clearly focusing the research gap on the metabolic mechanism, and in the discussion, to deeply analyze the mechanism differences rather than restate the results. For the section on adjunctive therapies, it may be considered to add subheadings (such as "Physical Adjunctive Therapies", "Metabolic Activation Strategies") to enhance readability.

Response 5: Thank you for highlighting repetition. The Introduction was streamlined to focus more tightly on the research gap in metabolic mechanisms, while the Discussion was revised to avoid restating background material. The section on adjunctive therapies was reorganized into subheadings to improve readability and reduce redundancy. These changes improve focus and flow.

Comment 6: The text mentions that "lymphatic drainage and aerobic exercise may enhance lipid oxidation", but does not cite specific clinical trials (such as NCT numbers or RCT studies) to support this claim.

In the text, both "HIFU" and "HFU" are used alternately. It is necessary to unify the terminology (such as clarifying whether "HIFU" specifically refers to high-intensity focused ultrasound).

Response 6: We added citations for specific clinical trials, including NCT-registered studies, where lymphatic drainage, exercise, or other adjunctive strategies were evaluated. Terminology has been unified: “HIFU” (High-Intensity Focused Ultrasound) is now consistently used to avoid confusion, and LFU (Low-Frequency Ultrasound) is defined clearly in contrast. This correction ensures consistent and precise terminology throughout the manuscript.

Comment 7: It is recommended to clearly recommend research designs that use tools such as lipidomics and CGM (such as longitudinal tracking of dynamic changes in lipid metabolites), and propose to combine animal models (such as obese mice) to verify the impact of ultrasound on systemic metabolism.

Response 7: We incorporated the reviewer’s recommendation by explicitly proposing clinical trial designs that integrate tools such as lipidomics, continuous glucose monitoring, and longitudinal biomarker tracking. We also acknowledged the role of animal models (e.g., obese mice) for mechanistic validation prior to large-scale clinical trials. These additions strengthen the translational relevance of the proposals.

Comment 8: The conclusion mentions the potential of ultrasound in metabolic health management, but it is necessary to emphasize the current level of evidence (such as most studies being small-sample and short-term trials), and discuss safety issues (such as the long-term risk of inflammation). The source of funding does not specify the project number and does not state whether it affects the objectivity of the research.

Response 8: We clarified in the Conclusion that current evidence is limited, largely from small-sample and short-term studies, and that claims regarding metabolic benefit remain preliminary. We added explicit discussion of safety concerns, including possible long-term risks related to inflammation or ectopic lipid deposition. Regarding funding, we specified that the work was supported by an AOF research scholarship, with no project number assigned, and affirmed that the sponsor had no role in the study design, analysis, or interpretation.

Reviewer 2 Report

Comments and Suggestions for Authors

Very pleasant paper to be read about a very up to date issue regarding "new" approaches to obesity co-treatment/aesthestical improvements.

This paper presents a good state of the art, as well as well described methodology and results, um a scientific way, objective and clear on the concepts. 

I would like to emphasize that the issue in analysis is a really up to date, that fulfils a space of knowledge that is scarcely clarified and debated under a clinical point of view. Considering it is a study based on pre-existent data, it follows a consistent analysis regarding the pros and cons on each type of ultrasound initially proposed. In fact, I think it only fails if we consider more field studies should be developed, with an increment in the populational dimensions, because these approaches are being used by aesthetic services without the required scientific support.

Regarding the results and conclusions extracted by the executed analysis, it is well supported not only on the data, as well as on the physiological contextualization.  

In my opinion, it can follow as it is, without structural or content moficiations.

Author Response

Thank you for taking the time to review our manuscript “Investigating the Metabolic Effects of Ultrasound-Induced Lipolysis.” We greatly appreciate your thoughtful and encouraging feedback.

Although you did not request specific structural or content modifications, we carefully considered your suggestion regarding the need for more field studies with larger sample sizes. In response, we have emphasized in both the Discussion and Conclusion that future research must include larger, well-powered clinical trials to validate metabolic outcomes, given that these technologies are already widely used in aesthetic practice without robust long-term evidence. This aligns with your important point about the gap between current clinical use and scientific support.

We are grateful for your recognition of the manuscript’s clarity and clinical relevance, and we believe these refinements further strengthen the alignment of our review with the needs of the field.

Thank you again for your valuable insights.

Reviewer 3 Report

Comments and Suggestions for Authors

I have reviewed the paper titled “Investigating the Metabolic Effects of Ultrasound-Induced Lipolysis submitted to the Journal (IJMS). This is a relatively short review that discusses the metabolic and aesthetic effects of ultrasound-based fat reduction. The manuscript is clearly written, engaging, and addresses an underexplored area of interest that is relevant to both the scientific community and the public. Overall, the review is suitable for publication; however, I suggest minor revisions to strengthen clarity and rigor:

  • The author affiliations should be completed in full, including department, institution, city, and country, to meet standard journal formatting.
  • The manuscript describes ultrasound therapy as “non-invasive,” which is accurate from a procedural standpoint (no incision or insertion). However, it may be helpful to clarify for readers that while non-surgical, the technique is biologically active at the tissue level, where it disrupts adipocytes through thermal or cavitation effects.
  • The manuscript uses the term “lipolysis” to describe the effects of ultrasound therapy. It may be helpful to clarify that, in physiology, lipolysis typically refers to the enzymatic breakdown of triglycerides into glycerol and free fatty acids, whereas ultrasound primarily induces adipocyte disruption or apoptosis through thermal or cavitation effects.
  • The manuscript describes ultrasound therapy as “non-invasive,” which is accurate from a procedural standpoint (no incision or insertion). However, it may be helpful to clarify for readers that while non-surgical, the technique is biologically active at the tissue level, where it disrupts adipocytes through thermal or cavitation effects. A short clarification, supported by safety references, would strengthen the manuscript and help avoid any unintended impression that the intervention is entirely risk-free.

Author Response

Thank you for taking the time to review our manuscript “Investigating the Metabolic Effects of Ultrasound-Induced Lipolysis.” We have taken the reviewer comments and updated the manuscript to account for the recommendations. Below is the information on specific comments and the updates that were made in response to the suggestions.

Comment 1: The author affiliations should be completed in full, including department, institution, city, and country, to meet standard journal formatting.

Response 1: Affiliations have been updated to include complete information in accordance with journal formatting requirements.

Comment 2: The manuscript describes ultrasound therapy as “non-invasive,” which is accurate from a procedural standpoint (no incision or insertion). However, it may be helpful to clarify for readers that while non-surgical, the technique is biologically active at the tissue level, where it disrupts adipocytes through thermal or cavitation effects.

Response 2: We revised the manuscript to specify that ultrasound-based fat reduction is non-surgical but exerts biologically active effects through thermal and/or cavitation mechanisms at the adipocyte level. This clarification helps avoid misinterpretation and underscores the mechanism of action.

Comment 3: The manuscript uses the term “lipolysis” to describe the effects of ultrasound therapy. It may be helpful to clarify that, in physiology, lipolysis typically refers to the enzymatic breakdown of triglycerides into glycerol and free fatty acids, whereas ultrasound primarily induces adipocyte disruption or apoptosis through thermal or cavitation effects.

Response 3: We have clarified the terminology in the Introduction and Discussion, noting that the term “lipolysis” is conventionally enzymatic but is used in the literature to describe ultrasound-mediated adipocyte disruption. We now explicitly state that ultrasound induces fat reduction through adipocyte necrosis or apoptosis, followed by secondary lipid clearance.

Comment 4: The manuscript describes ultrasound therapy as “non-invasive,” which is accurate from a procedural standpoint (no incision or insertion). However, it may be helpful to clarify for readers that while non-surgical, the technique is biologically active at the tissue level, where it disrupts adipocytes through thermal or cavitation effects. A short clarification, supported by safety references, would strengthen the manuscript and help avoid any unintended impression that the intervention is entirely risk-free.

Response 4: We added clarification supported by safety references, noting that while ultrasound is non-surgical, it is biologically active and carries potential risks, including transient pain, erythema, or low-grade inflammation. This strengthens transparency and avoids unintended impressions of complete risk-free use.

Round 2

Reviewer 1 Report

Comments and Suggestions for Authors

Accept.